# Surface Modification of Fluorescent Nanodiamonds for Biological Applications

**DOI:** 10.3390/nano11010153

**Published:** 2021-01-09

**Authors:** Hak-Sung Jung, Keir C. Neuman

**Affiliations:** Laboratory of Single Molecule Biophysics, National Heart, Lung and Blood Institute, National Institutes of Health, Bethesda, MD 20892, USA; haksung.jung@nih.gov

**Keywords:** fluorescent nanodiamond, nitrogen vacancy center, detonation nanodiamond, biocompatibility, functionalization, biological applications

## Abstract

Fluorescent nanodiamonds (FNDs) are a new class of carbon nanomaterials that offer great promise for biological applications such as cell labeling, imaging, and sensing due to their exceptional optical properties and biocompatibility. Implementation of these applications requires reliable and precise surface functionalization. Although diamonds are generally considered inert, they typically possess diverse surface groups that permit a range of different functionalization strategies. This review provides an overview of nanodiamond surface functionalization methods including homogeneous surface termination approaches (hydrogenation, halogenation, amination, oxidation, and reduction), in addition to covalent and non-covalent surface modification with different functional moieties. Furthermore, the subsequent coupling of biomolecules onto functionalized nanodiamonds is reviewed. Finally, biomedical applications of nanodiamonds are discussed in the context of functionalization.

## 1. Introduction

Among carbon nanomaterials, nanosized diamond particles or nanodiamonds (NDs), including detonation nanodiamonds (DNDs) and nanodiamonds containing fluorescent color centers, termed fluorescent nanodiamonds (FNDs), have received increasing interest. In particular, due to their high biocompatibility and unique optical properties including extraordinary photostability (neither photobleaching nor blinking), near infrared (NIR) fluorescence emission (≈650–900 nm), long fluorescence lifetime (≈20 ns) and high fluorescence quantum yield, FNDs containing nitrogen-vacancy (NV) centers are being used in an expanding number of biomedical applications [1,2,3,4,5,6]. Nitrogen, substituted in place of a carbon atom, is the most common impurity in diamonds. Natural diamonds frequently contain nitrogen atom impurities, and nitrogen is often doped into synthetic diamonds [7]. NV centers, which are localized defects comprising a lattice vacancy adjacent to a substitutional nitrogen in the crystalline diamond lattice, are generated via irradiation and thermal treatment of NDs [8]. NV centers imbue FNDs with exceptional optical properties, which, combined with the exceptional mechanical and chemical stability of diamonds, result in versatile and robust fluorescent nanoparticles. NV centers can be statistically formed with two different charged states: neutral (NV^0^, electron spin: *S* = 1/2) or negative (NV^−^, *S* = 1), with zero-phonon lines at 575 and 637 nm, respectively [9,10]. The charge state of an NV center is not fixed but rather can transition between the NV^0^ and NV^−^ states in response to functional groups and impurities on the diamond surface [11]. This process has been modeled as an equilibrium between electron donors and the NV center, which can be shifted by the presence of electron traps or donors on the diamond surface [11]. Consistent with this model, NV^−^ centers are more prevalent in oxidized, negatively charged NDs [12,13]. Generally speaking, functionalization schemes that eliminate surface impurities such as graphitic carbon, and that leave the FND surface negatively charged will favor the formation and stabilization of the more useful NV^−^ charge state. Another interpretation is that the charge state can be reversibly switched depending on the position of the NV^0^ and NV^−^ ground state levels with respect to the Fermi level at the FND surface [11,14,15]. In other words, an electron can be eliminated from the NV^−^ center when its ground state level is moved below the Fermi level and an electron will be added to the NV^0^ center when its ground state level is shifted above the Fermi level. This surface-charge-induced modulation of fluorescence was exploited to monitor noncovalent surface chemical processes and molecular binding events that altered the charge density in proximity to the ND surface [15].

The electron spin transitions of NV^−^ centers can be manipulated using microwave radiation at ~2.87 GHz, which results in changes in their emission intensity that can be further modulated by an external magnetic field [1]. This phenomenon has been exploited to provide an optical readout of spin state, which has been termed optically detected magnetic resonance (ODMR). ODMR with FNDs has been harnessed to make exquisitely sensitive measurements of magnetic and electric fields and, more recently, of temperature at the nanoscale [16,17,18,19]. Owing to the accommodation of other color centers such as N3, H3, SiV, GeV, and Ni in diamond lattice structure, the emission spectra of FNDs can cover a wide range of the Vis to NIR spectrum [20,21,22,23]. Consequently, FNDs have been used in diverse biomedical fields such as imaging, tracking, sensing, and drug delivery due to their excellent physicochemical properties and commercial availability [24,25,26,27].

NDs can be directly obtained from a detonation process (DNDs), or through mechanical size reduction of high-pressure high-temperature (HPHT) diamonds [7,28,29]. The detonation method employs explosives that provide both a carbon source and the requisite energy to form diamonds when detonated in an inert atmosphere in a metallic chamber [28,30]. The resulting DNDs have an average diameter of 4‒5 nm. Since DNDs were developed before other nanodiamond forms and have been commercially available for some time, the majority of purification, functionalization, and applications have been developed for DNDs [30,31]. The HPHT method produces large diamonds that are subsequently irradiated with high energy electrons, protons, or helium ion to create NV centers, followed by mechanical size reduction and selection to generate FNDs [29,32,33,34].

Since requirements including high dispersibility, conjugation with functional molecules, and biocompatibility can be satisfied by appropriate functionalization of the FND surface, the range and specificity of applications involving FNDs is quite large and continues to expand as new functionalization approaches are developed. For example, in drug delivery applications, FND surface functionalization strategies can be roughly classified as passive or active targeting [35,36,37,38]. Passive targeting is achieved by controlling the size and surface properties such as charge and hydrophobicity of the FNDs. Even though passive targeting improves pharmacokinetics, selectivity, and reduces side effects, it has faced several challenges including difficulties with process control, high sensitivity to environmental factors, and nonspecific accumulation [39,40]. Active targeting was devised as a complementary method to passive targeting. It is achieved by surface functionalization of the FND with molecules that specifically and tightly bind the biological target, avoiding nonspecific biding to other cells, tissues, and organs. The resulting improvements in selectivity by specific binding of target sites and reduction of non-specific interactions with normal tissues, can increase therapeutic efficiency and reduce side effects [41,42]. Hence, surface modification and specific molecular conjugation approaches for FNDs are of paramount importance for developing targeted delivery of FNDs for biomedical applications.

The majority of ND surface modification approaches have been developed for DNDs due to the early development this synthesis method and the fact that they are commercially available [30,31]. Although there are differences among the surface moieties found on DNDs and FNDs derived from other sources (typically milling or other mechanical size reduction of synthetic diamonds), homogenization and further chemical reactions developed for DNDs can generally be applied to FND surfaces [43]. Despite differences in synthesis and vacancy formation between DNDs and HPHT nanodiamonds, the distinction between them is gradually blurring due to the continued development of synthetic methods and purification approaches. For example, NV color centers have been created in DNDs by electron irradiation, and the size of HPHT particles can be reduced to below several nanometers, depending on the synthesis and size reduction strategies [44,45,46]. In addition to size, sp^2^ carbon, abundant on DNDs, has typically differentiated DNDs and HPHT particles. However, sp^2^ carbon is routinely found on HPHT nanodiamond surfaces, likely due to irradiation and post-irradiation processing steps [11]. Methods have been developed to remove sp^2^ from both DNDs and HPHT nanodiamonds, which results in similar, if not identical, sp^3^ carbon forms for both particles [46,47,48,49]. Therefore, many of the functionalization strategies originally developed for DNDs have been successfully adapted for use with HPHT nanodiamonds [8,50,51,52]. Generally, a large number of chemical groups can be generated on the ND surface during production, with sp^2^ carbon and metal impurities including graphene-like or graphitic structures being the most common, though the amount and nature of impurities depend on the details of the production process [53]. To remove unwanted carbon structures and to homogenize the surface groups on NDs, several oxidation or reduction strategies have been developed [54,55,56,57,58]. Thus, the final surface functionality of NDs is determined by the production method and subsequent purification and modification steps. NDs can be further functionalized after homogenization via covalent and non-covalent surface chemistries. Due to the blurring of the sharp distinction among NDs, DNDs, and FNDs, the review covers functionalization approaches applied to nanodiamonds. Approaches that have been specifically applied to FNDs are indicated, though we anticipate that the vast majority, if not all, of the functionalization schemes would be suitable and applicable for FNDs.

## 2. Formation of Uniform Surface Moieties on NDs

Pristine NDs have various surface moieties, as well as nondiamond carbon structures and impurities that are generated during synthesis [59,60]. Nondiamond carbon structures such as carbon dots can affect the fluorescence properties of NDs [34]. Thus, elimination of impurities and homogenization of the surface groups are required prior to performing further reactions on the surface of NDs to increase the purity, to control subsequent reactions on the ND surfaces, and to maximize the contribution of NV centers to the optical properties of FNDs. Consequentially, numerous studies have established homogenization procedures for NDs, such as carboxylation, hydroxylation, hydrogenation, halogenation, amination, and graphitization [54,55,56,57,58].

### 2.1. Carboxyl Groups

During production of NDs, sp^2^ carbon structures and metallic impurities contaminate the surface of the NDs [53,59,60]. To remove sp^2^ carbon and other surface impurities, oxidation treatment with mineral acids and air or ozonized air is carried out, which ideally results in the formation of carboxyl COOH groups on the ND surface (Figure 1A) [54]. Carboxylation has attracted a great deal of attention due to the wide range of versatile secondary reactions possible with COOH groups. A mixture of several different acids such as nitric (HNO_3_), sulfuric (H_2_SO_4_), hydrochloric (HCl) and perchloric acid (HClO_4_) with additives including potassium dichromate (K_2_Cr_2_O_7_) and potassium nitrate (KNO_3_), are used to purify NDs on an industrial scale [54,60,61]. In addition, a mixture of H_2_SO_4_, concentrated hydrogen peroxide (H_2_O_2_), and supercritical water can be employed to purify NDs [11,62]. There is an alternate strategy for removing impurities in a more environmentally friendly and cost-effective manner by air and/or ozone treatment at elevated temperatures [59,63]. This technique significantly increases the purity of the NDs and generates FNDs with improved dispersity in solvents. In general, the oxidation reactivity of disordered sp^2^ carbon is significantly higher than that of diamond. Hence, the oxidation of NDs improves the phase purity of the carbonaceous material and introduces oxygenated groups on the surface. Furthermore, the generated COOH groups on the ND surface provide several facile chemistries to functionalize NDs for further applications.

### 2.2. Hydroxylation

Owing to the possibility of a broad range of subsequent reactions of hydroxyl (OH) groups, the homogenization of the ND surface with dense OH groups is one of the most common homogenization schemes. Hydroxylation of NDs is carried out four different ways; through reduction with borane (BH_3_·THF) or Lithium aluminum hydride (LiAlH_4_), via a Fenton reaction (a solution of H_2_O_2_ with iron (II) sulfate (FeSO_4_)), mechanochemical treatment, or through a photochemical reaction (Figure 1B) [48,58,64,65,66]. Although the oxygen containing surface groups including anhydrides, carbonyls and aldehydes are nominally converted to OH groups by reduction with BH_3_·THF, the conversion efficiency depends on the identity of the initial surface groups on the ND. Therefore, LiAlH_4_ is alternatively used to reduce all acid derivatives to alcohol moieties. However, sp^2^ carbon structures cannot be removed and side products including aluminum oxide hydroxides can be generated by LiAlH_4_. The Fenton reaction also generates OH groups on the ND surface with high efficiency. Not only is sp^2^ carbon removed by oxidation but also the hydroxyl radicals (·OH) generated in situ react with the ND surface to generate hydroxylated ND. The mechanochemical treatment of NDs in water (H_2_O) is one of the simplest methods to produce a considerable density of OH groups on the ND surface. The formation of OH groups on the ND surface is thought to occur through nucleophilic attack on free spins generated during milling by water molecules in the slurry or in the washing media. Finally, atomic oxygen and OH radicals produced by UV irradiation in the presence of O_2_ and H_2_O results in the generation of terminal OH groups on the ND surface with concomitant elimination of other oxygenated surface moieties. Recently, it was demonstrated that hydrogen atoms on hydrogenated NDs can be directly converted to hydroxyl groups by atmospheric pressure radio-frequency micro-plasma jet treatment [67].

### 2.3. Hydrogenation

Hydrogenation of the ND surface can be accomplished via thermal annealing or microwave plasma treatment in a hydrogen atmosphere (Figure 1C) [55,68,69]. The hydrogenation reaction mechanism is thought to include decomposition, decarbonylation and/or decarboxylation as the main reactions taking place upon treatment of NDs with mixtures of hydrogen and nitrogen. Despite the simplicity and potential utility of this approach, concerns remain regarding the reaction conditions and possible generation of other moieties including CH_2_, CH_3_, and OH on the ND surface. These concerns limit the applicability of hydrogenation, which would benefit from the development of new strategies with mild reaction conditions that nonetheless achieve complete conversion of surface groups to hydrogen on the ND surface.

### 2.4. Amination

Amino groups permit the most versatile reaction for the surface modification of nanoparticles through either the formation of amides using many established protocols or reductive amination. Thus, several approaches to graft NH_2_ groups on ND surfaces have been developed including ultraviolet (UV) irradiation of hydrogenated diamonds in the presence of chlorine gas, and treatment of chlorinated NDs with ammonia at elevated temperatures (Figure 1D) [70]. Amination of NDs can also be achieved via further covalent or non-covalent reactions with hydrogenated, carboxylated, or hydroxylated NDs. In one scheme, OH groups on the ND surface are treated with (3-aminopropyl)trimethoxysilane (APTES) that results in the generation of an amino group linked to the surface via the APTES molecule [58]. Alternatively, hydrogenated NDs are derivatized with aryl diazonium to form a 4-nitrophenyl coupled ND complex, and amine groups are subsequently formed by electrochemical reduction of the 4-nitrophenyl functionalized ND [71]. In a non-covalent approach to introduce amino groups to the surface of NDs, poly-L-lysine is physically absorbed to the carboxylated and oxidized ND surface via electrostatic and hydrogen bonding [72].

### 2.5. Halogenation

Halogenation of NDs is a convenient method of surface activation via the creation of electrophilic centers capable of further reaction with multiple nucleophilic reagents (Figure 1E). To create fluorinated NDs, F_2_/H_2_ mixture gases are introduced to pristine NDs at elevated temperatures [57]. Alternatively, sulfur hexafluoride (SF_6_) with atmospheric pressure plasma can be applied to introduce C–F bonds on the ND surface [73]. Recently, fluorination of carboxylated NDs was achieved via silver (Ag)-catalyzed selective substitution of surface carboxyl groups to fluorine using the Selectfluor reagent under mild reaction conditions [74]. The photochemical reaction of hydrogenated NDs with gas phase chlorine has also been employed to create chlorinated NDs [70]. Another method for creating C–Cl bonds on ND surfaces involves treatment with dry chlorine or CCl_4_ vapor at high temperatures [75]. A variant of this approach, liquid-phase chlorination of NDs by molecular chlorine, has been proposed [76]. Alternatively, chlorination of ND surfaces has been carried out via an electron beam irradiation induced radical addition mechanism [77]. Considerable bromination of NDs has been demonstrated using *N*-bromosuccinimide (NBS) as a bromine source with hydroxylated NDs in dry CCl_4_ under mild reaction conditions [65].

## 3. Non-Covalent Functionalization

Although the properties of NDs open up numerous possible applications, the feasibility of implementing these applications is limited by the inherent surface properties of NDs as well as aggregation in physiological conditions. Consequently, surface modification of NDs to increase colloidal stability and functionality is one of the most important prerequisites for many applications. Functionalization schemes of NDs are generally classified as non-covalent or covalent (Figure 2 and Figure 3). The diversity of NDs’ surface chemical groups achieved via homogenization processes enable conjugation of various functional materials including DNA, antigens, proteins, peptides, fluorescent dye molecules, and drugs through conventional organic chemistry. A comparatively simple and flexible strategy for modifying the ND surface with desired functional molecules through physical adsorption is non-covalent functionalization.

Generally, NDs purified with mineral acids exhibit a hydrophilic surface and negative surface charge with a high density of different oxygen-containing surface moieties. These surface groups allow several different interactions (e.g., hydrogen bonding, electrostatic, and Van der Waals interaction ≈ physical adsorption) with functional molecules (Figure 2). For example, bovine insulin, apoobelin, lysozyme, and doxorubicin hydrochloride have been directly immobilized on ND surfaces via physical adsorption [79,88,89,90]. However, the activity or availability of non-specifically adsorbed molecules on the ND surface can be influenced by the size of the ND [91]. The strong interaction between small sized NDs (5–50 nm) and attached molecules can result in structural changes or unfavorable orientations of the molecules on the surface of the NDs. In these cases, the interaction between the selective binding site of the functional molecule on the ND surface and specific reactive surface on the cognate ligand may be disturbed. Alternatively, positively charged polymers can be attached on the ND surface via physical adsorption. The introduced polymers provide multiple bonds and/or interactions with the ND surface as well as effective functional groups for subsequent reactions to immobilize biomolecules and drugs through various chemical reactions or interactions (Figure 2A) [36,72,78]. Positively charged polymers such as poly-L-lysine, polyarginine, and polyethyleneimine have been used to functionalize the ND surface with a high number of primary amino groups, which can then form amide bonds with other functional moieties through amine chemistry. In one example, the hormone insulin was physically adsorbed on ND surfaces for drug delivery applications (Figure 2B). Since insulin is relatively stable and produces a measurable cellular response, it is used as a model for protein desorption. Considering the isoelectric point and amine groups of insulin, electrostatic interactions and H-bonding likely dominate its surface adsorption on the NDs. These interactions result in both high loading capacity (80%) of insulin onto NDs, as well as pH-mediated desorption (up to 46%) [79]. The steroid receptor-targeted magnetic resonance (MR) imaging contrast agent, ProGlo, has limitations due to its hydrophobicity [80]. To overcome this limitation, ProGlo was physically adsorbed onto NDs. Due to the resulting enhanced water solubility, the ProGlo conjugated NDs afforded high MR contrast efficiency (Figure 2C). In another example, PEG and the abundant blood plasma protein human serum albumin (HSA) were connected by a sequential cationization and *N*-hydroxysuccinimide ester reaction. The HSA-PEG was further functionalized with doxorubicin (DOX) and the resulting DOX conjugated cationic HSA-PEG was non-specifically adsorbed onto NDs [36]. Although the functionalized NDs contained a large number of hydrophobic DOX molecules, they retained their colloidal stability in physiological and highly concentrated salt solutions. Due to this high colloidal stability, the DOX nanodiamonds exhibited high biocompatibility as well as extended circulation in the blood stream, resulting in efficient delivery of high concentrations of DOX molecules to cancer cells.

## 4. Covalent Functionalization

Despite the simplicity of non-covalent functionalization approaches, they can reduce the activity of bioactive molecules due to multiple bonding between the functional molecule and the ND surface [91]. Furthermore, the density of non-covalent immobilized molecules on the ND surface is relatively low compared to covalent attachment schemes [92]. These limitations of non-covalent functionalization schemes can be solved by covalent functionalization approaches. Indeed, the biocompatibility of NDs can be enhanced by covalent surface functionalization with polyethylene glycol (PEG) or poly glycerol (PG). Depending on the original ND surface groups, different covalent surface modification methods such as esterification, amidation, nucleophilic reaction, and silanization can be employed.

### 4.1. Carboxyl Groups

COOH groups on the NDs surface provide one of the most reliable functionalization strategies through the formation of amide bonds via covalent conjugation of amines (Figure 3A). The formation of amide bonds can be achieved under mild reaction conditions (in water at room temperature) with well-established and robust 1-ethyl-3-(3-dimethylaminopropyl)carbodiimide (EDC) or *N*-hydroxysuccinimide (NHS) chemistry [81,93]. Therefore, the reaction conditions to form amide bonds are suitable for covalent conjugation of different hydrophilic materials to the ND surface. Because the EDC coupling reaction is straightforward, reliable, and efficient, it is suitable for many ND applications requiring surface functionalization [94,95,96]. The activation of COOH with thionyl chloride (SOCl_2_) generates acyl chloride, which can directly react with amines to produce the amide [82,93]. This reaction can also be employed to generate amine terminated NDs by reacting diamines with acyl functionalized NDs [97]. The COOH groups on the ND surface can also be transformed to azide groups via Ag(I)-mediated decarboxylation. The converted azide provides a convenient approach for further functionalization via click chemistry [98]. Alternatively, COOH groups can readily form ester bonds with appropriate alcohols [93]. The ester bonds are generated either in the presence of an acid catalyst or via the significantly more reactive acid chloride. However, ester bonds tend to spontaneously hydrolyze in aqueous environments because the chemical stability of an ester bond is relatively low compared to an amide bond.

### 4.2. Hydroxyl Groups

OH groups on the ND surface can be employed for functionalization via several chemical reactions including formation of ester or ether bonds, silanization, and polymerization (Figure 3B) [58,83,84,85]. Ether bonds are formed by reaction of hydroxylated NDs with alkyl chlorides in the presence of sodium hydride (NaH) [83]. Since NaH is a strong base, it deprotonates OH groups on the ND surface. The deprotonated NDs act as nucleophiles that attack the alkyl halide to produce an ether linkage. Alternatively, OH groups on the ND surface enable the formation of an ester via reaction with SOCl_2_ activated COOH. However, as mentioned above, ND-esters of alkyl carboxylic acids are not stable in water. Hence ester linked NDs can be converted to hydroxylated ND when dispersed in water due to hydrolysis of the ester bond on the ND surface. Hydroxylated NDs can also be covalently modified with organotrialkoxysilane (R–Si(OR’)_3_) through a silanization reaction, and the terminal functional groups (R) can be subsequently functionalized [58]. Although the properties of OH groups between silica and ND surface are different, there is a concern that the bond linking organotrialkoxysilane molecules to the hydroxyl group on silica surfaces can be reversed by hydrolysis [99]. In particular, the widely used aminofunctional organotrialkoxysilane, (3-aminopropyl)trimethoxysilane (APS), tends to reduce colloidal stability in water and physiological buffers due to low hydrolytic stability [100,101].

The OH groups on the ND surface can also be transformed into other functional groups such as NH_2_, Cl or Br [58,65]. In another innovative approach, dopamine derivatives bearing terminal azide (N_3_) groups reacted with hydroxylated NDs through the catechol anchor of dopamine, permitting subsequent click chemistry of the azide with an alkynyl-pyrene fluorescent molecule in the presence of copper(I)-catalyst [102].

### 4.3. Halogen Groups

Halogen atoms are good leaving groups. Therefore, the bond between the carbon and the halogen is readily cleaved in the presence of a sufficiently nucleophilic reagent via the substitution reaction. Halogenated NDs have been produced by activation with electrophilic centers for further reaction with a variety of nucleophilic reagents [57]. For example, fluorinated NDs are reacted with nucleophilic reagents (alkyllithium compounds), diamines, and amino acids by intermolecular elimination of LiF or HF during the reaction (Figure 3C). Chlorinated NDs also enable reactions with different nucleophiles including organolithium reagents and cyanide ions [76].

### 4.4. Amine Groups

There are numerous established conjugation strategies to form chemical bonds with primary amines. These include NHS esters, sulfonyl chlorides, aldehydes, isocyanates, acyl azides, epoxides, oxiranes, carbonates, aryl halides, imidoesters, carbodiimides, anhydrides, and fluorophenyl esters [77]. Aminated NDs were conjugated to both a fluorescent dye and a biotin moiety using their NHS derivatives (Figure 3D) [64]. In another coupling scheme, the NH_2_ groups on aminated NDs were substituted to N_3_ through the reaction with 4-azidobenzoic acid in the presence of a carbodiimide coupling agent [103]. A variety of alkyne-containing compounds, including decyne, ethynylferrocene, and *N*-propargyl-1-pyrenecarboxamide, can be subsequently coupled to the ND-N_3_ using click chemistry.

### 4.5. Hydrogen Groups

Hydrogenated NDs represent a convenient starting point for functionalization due to the homogeneous surface chemistry, specific subsequent reactions, and robust carbon–carbon single bonds (C–C bond) at the interface that are resistant to oxidation and hydrolysis (Figure 3E) [70,71,104]. The functionalization of hydrogenated NDs is carried out via a radical reaction with carboxylic acid to conjugate *ω*-amino acids through an ester linkage [105]. 4-bromophenyldiazonium tetrafluoroborate is used to functionalize the hydrogenated ND surface and the generated 4-bromophenyl is further functionalized via the Suzuki coupling reaction [71]. The diazonium coupling reaction on hydrogenated NDs is achieved by immersing the ND-H into a solution containing aryldiazonium salts. Heteroatom bonded linkers such as O, N, and S can result in lower stability between the ND surface and the attached functional molecules. To solve this problem, the C–C bond can be formed through a photochemical process [104]. In this scheme, 500 nm hydrogenated NDs were functionalized with an optically active amide under irradiation with a low-pressure mercury lamp.

### 4.6. Polymer Coating

To increase colloidal stability and blood circulation time, and reduce non-specific adsorption of proteins, and biocompatibility of ND for biomedical applications, polymers are introduced onto the ND surface. PEG and PG represent two different polymer grafting approaches as well as two polymers commonly used to stabilize NDs [86]. Polymer coating of NDs can be achieved via non-covalent (physical adsorption) or specific covalent attachment (amide or ester linkages) via monomer addition or fully synthesized polymers. This ND functionalization strategy can be divided into “grafted to” and “grafted from” (Figure 3F) [86,87]. Fully synthesized polymers are used in the “grafted to” approach to modify the ND surface. Diverse functional groups (amine, thiol, hydroxyl, methacrylate, aldehyde, and NHS) terminated PEG molecules have been used [87,95,106,107,108,109,110]. A commonly employed scheme to attach PEG to the ND surface uses carboxylated NDs conjugated to amine functionalized PEG via a standard EDC coupling reaction [111]. PEG functionalized NDs formed via this EDC coupling reaction have been validated for multi-drug loading via physical adsorption [95]. On the other hand, heterobifunctional PEG bearing NHS at one end of the back bone and alkyne moiety at the other can be conjugated to amine terminated NDs through an EDC coupling reaction with the NHS group. The alkyne group permits further conjugation of various molecules to the ND by click chemistry [109]. This is an attractive and versatile approach, since the hydrophilic PEG between the ND surface and the functional biomolecule increases the mobility of the attached biomolecule, thereby increasing the probability that it retains activity and stability in aqueous solution. Although PEG exhibits many advantages, it can generate aldehydes under in vivo conditions through enzyme-catalyzed oxidation of hydroxyl groups on the PEG chain as well as immune reactions associated with binding of the PEG-specific immunoglobulin M [112,113,114,115]. PG has been shown to overcome many of the drawbacks of PEG. PG can be polymerized on the surface of hydroxylated and/or carboxylated NDs through ring-opening multi-branching polymerization of glycidol [25,26,116,117]. This type of reaction is termed “grafted from”. PG-modified NDs exhibit good solubility in a variety of buffers and methanol. Owing to decreased nonspecific adsorption, PG-functionalized NDs showed much higher tumor accumulation [117]. In addition to PG, poly(oligo(ethylene glycol) methyl ether methacrylate), poly(*N*-isopropylacrylamide), and poly(2-methacryloyloxyethyl phosphorylcholine) have been employed to functionalize NDs via reversible addition−fragmentation chain transfer, free radical, and metal-free atom transfer radical polymerization in the “grafted from” approach, respectively [118,119,120]. Since the “grafted from” method is free from steric hinderance associated with long polymers, a higher density of polymer molecules can be conjugated to the ND surface compared to “grafted to” or non-covalent polymer coating approaches [87].

## 5. Combination of Non-Covalent and Covalent Functionalization

Each functionalization scheme (covalent and non-covalent) has its own pros and cons. Encapsulation approaches compensate for disadvantages, while maintaining the benefits, of each strategy. Encapsulation of NDs is employed to both increase colloidal stability and to facilitate further bioconjugation. Since the late 1960s, the Stöber method has been widely used to grow monodisperse spherical silica nanoparticles (~100 nm) based on the sol–gel chemistry of silicon alkoxides in basic aqueous solutions containing different alcohols such as methanol, ethanol, or isopropanol [121]. Adopting the Stöber method, amorphous silica has been used as a conventional coating material for numerous nanoparticles due to its exceptional stability, easy regulation of the coating process, chemical inertness, controlled porosity, processability, optical transparency, and biocompatibility as well as well-established silane chemistry to introduce silanol groups on the silica shell [122]. To generate a silica shell on the ND surface, NDs in a solution of tetraethyl orthosilicate (TEOS) were confined in 1-palmitoyl-2-oleoyl-*sn*-glycero-3-phosphocholine multilamellar vesicles, which break into small unilamellar vesicles under ultrasonication with nominal diameters of ~100 nm (Figure 4A). The trapped TEOS is converted into silica on the ND surface by hydrolysis and condensation reactions with triethylamine base catalyst. The large number of silanol groups on the surface of silica encapsulated ND were subsequently reacted with APTES as an intermediate linker for conjugation of an amine-reactive biotin moiety [123]. A silica shell can be also grown in the presence of polyvinylpyrrolidone. To improve the resistance of silica shell against hydrolysis or to permit secondary reactions, a crosslinked amino-functionalized layer can be formed on the silica shell with a mixture of APTES and bis(triethoxysilyl)ethane. These Amino-functionalized silica shell encapsulated NDs were conjugated with a heterobifunctional PEG chain bearing an *N*-hydroxysuccinimidyl group at one end of the chain and an alkyne moiety at the other. The alkyne moiety provides a facile approach to attach various molecules to the particle via click chemistry [109]. Mesoporous silica shells can also be grown on the ND surface using cetyltrimethylammonium bromide as a pore-generating agent. The mesoporous silica coated NDs were developed for drug delivery applications due to the high drug loading capacity afforded by the porosity of the silica shell [124]. Rapid and efficient encapsulation of NDs with polydopamine (PDA) shells has also been demonstrated [110]. Dopamine molecules undergo self-polymerization under mild basic conditions and the resulting PDA can deposit onto inorganic and organic surfaces including noble metals, oxides, polymers, semiconductors, and ceramics [125]. Thus, the PDA shell is readily generated on the ND surface by oxidation and self-polymerization of dopamine hydrochloride in a mild basic aqueous solution. Neither processing history, surface functionality, nor size influence the PDA encapsulation method. Furthermore, due to the abundant catechol/quinine groups on the surface of the PDA shell, it can be functionalized with thiol/amine containing molecules via Michael addition or Schiff base reactions under oxidizing conditions (Figure 4B). However, PDA can quench fluorescence and absorbs broadly in the UV and visible spectrum [126,127]. Consequently, the PDA shell thickness was controlled through reaction conditions to reduce quenching and optical absorption of the NV fluorescence. The PL intensity of PDA coated FND increased as the PDA shell thickness decreased. NDs have also been encapsulated in lipid membranes by photoinitiated radical polymerization of 1,2-bis(10,12-tricosadiynoyl)-*sn*-*glycero*-3-phosphocholine [128]. Since COOH groups can be uniformly introduced into the lipid shell, anti-CD44 antibody could subsequently be conjugated to the lipid-encapsulated NDs via carbodiimide chemistry (Figure 4C). Alternatively, both a thin-film hydration technique and a spontaneous emulsification scheme have been demonstrated for lipid encapsulation of NDs [129,130]. Due to the high structural stability of the lipid encapsulation, lipid-coated NDs showed high colloidal stability. In addition to their stability, lipid coated NDs were conjugated with specific antibodies and the functionalized particles were used as drug delivery vehicles and fluorescent fiducial markers.

## 6. Biological Applications of FNDs

FNDs have attracted a great deal of attention in biological fields such as imaging, drug delivery, and sensing applications due to the unique combination of optical and physicochemical properties along with the accessibility of numerous surface chemistries [4,5,6,131]. The intrinsic high photostability of color centers in the diamond lattice structure make FNDs ideal particles for long-term cell tracking, super resolution imaging, and correlative light- and electron- microscopy [130,132,133]. Furthermore, NV^−^ centers (total spin *S* = 1) with two unpaired electrons can be spin-polarized by optical pumping [134]. The spin levels of the electrons result in triplet ground (^3^A_2_) and excited (^3^E) states. In the absence of external fields, the triplet ground state *m_s_* = 0 and *m_s_* = ± 1 levels are split by an energy equal to the axial zero-field splitting (ZFS) parameter, D ≈ 2.87 GHz due to the spin-spin interactions between the unpaired electrons (Figure 5B) [1]. A resonant microwave field induces magnetic dipole transitions between the *m_s_* = 0 and *m_s_* = 1 states in the ground or excited states, which interferes with the optically pumped spin polarization, resulting in a decrease in the fluorescence intensity of the NV^−^ centers. This optical detection of the spin state of electrons in NV^−^ centers is called optically detected magnetic resonance (ODMR) [1]. This unique property enables background-free imaging and sensitive FND fluorescence detection by combining the selective modulation of FND fluorescence, via a resonant microwave field, with various phase sensitive detection techniques. In one approach, FNDs were tracked in both *Caenorhabditis elegans* and mice by subtraction of images with and without microwave irradiation to overcome background fluorescence [135]. In a similar approach, modest magnetic fields (greater than ~100 G) mix the *m_s_* = 0 and *m_s_* = 1 states and results in a decrease in the fluorescent intensity of the NV centers. Magnetic modulation of FND fluorescence intensity permitted background free imaging of FNDs in the lymph nodes of mice [136,137]. More recently, a microwave modulation approach was developed to improve the detection sensitivity of an in vitro diagnostic assay based on the capture and detection of nanoparticles. Selectively detecting the modulated FND fluorescence via lock-in techniques resulted in an improved detection limit of 8.2 × 10^−19^ molar for biotin–avidin, a 105-fold increase in sensitivity over gold nanoparticles, which are typically used for these assays [138]. The long fluorescence lifetime of NV centers permits an alternative background free imaging approach based on time gated imaging to distinguish FND emission from autofluorescence. In one implementation of this approach, FNDs non-specifically coated with *Caenorhabditis elegans* yolk lipoprotein complexes were tracked in vivo over the course of 50 min as they were transported to oocytes and incorporated into developing embryos [139]. Long-term, background-free imaging of FNDs in organisms opens up exciting possibilities in several biomedical fields. While FNDs are almost perfect imaging probes, they have emerged as useful drug delivery vehicles and offer a promising avenue to realize theranostic applications combining diagnostic and therapeutic functions.

The biocompatibility and facile surface modification of FNDs have been exploited for drug delivery applications. Notable examples include the development of an RDG-peptide targeted delivery platform for a platinum compound. In this approach, 50 nm FNDs with carboxylic acid surface groups were first stabilized with multibranched polyglycerol covalently linked via ring-opening polymerization of glycidol. Hydroxyl groups of the polyglycerol were subsequently used to conjugate a platinum-based drug and cell receptor targeting RGD peptide. These particles were selectively taken up by, and killed, cells expressing the RGD receptor [27]. In a similar approach, positively charged blood plasma protein human serum albumin, covalently conjugated with PEG and the anticancer agent doxorubicin, was non-specifically adsorbed onto the negatively charged FND surface. The functionalized FNDs exhibited high colloidal stability and biocompatibility. More importantly, they elicited a strong cell killing response and were shown to have potent anti-tumor activity in a xenograft breast cancer model, thus establishing the approach as a promising drug delivery and possible theranostic tool [36].

In addition to imaging and drug delivery or theranostic nanoparticles, the properties of NV^−^ centers within FNDs are promising for sensing applications in biological environments due to their biocompatibility, exceptional optical properties, and high magnetic sensitivity with nanoscale spatial resolution under ambient conditions (Figure 5A) [140]. NV^−^ centers in different diamond structures (nano- or microparticles, films) have been developed as nanoscale sensors due to their spin-dependent optical properties (ODMR) with high sensitivity for external perturbations including electric or magnetic field, temperature, and pressure [17,18,141,142,143,144]. Based on the temperature dependence of the zero-field splitting measured through ODMR, NV^−^ centers demonstrate intriguing potential for nanoscale thermometry within cells (Figure 5C). Coherent manipulation of the electronic spin associated with NV^−^ centers in diamond was harnessed in an innovative experiment in which temperature was simultaneously controlled and mapped at the subcellular level by introducing both FNDs and gold nanoparticles into a human embryonic fibroblast cell [145]. In addition to ODMR, different FND-based sensing methods, including spin coherence, spin relaxation time, spectral shift of the zero-phonon lines, and charge-state, have been proposed to measure the concentration of paramagnetic ions and molecules, temperature, and pH [50,51,52,146,147]. The presence of paramagnetic ions or proteins can influence the spin relaxation times, both longitudinal (*T_1_*) and transverse (*T_2_*) spin relaxation, of the NV^−^ center, and fluctuations of the electron spin of the NV^−^ directly respond to magnetic noise. Therefore, the concentration of paramagnetic species can be detected by measuring the fluctuations of NV^−^ spin relaxation times [146,147]. FNDs have also been adapted to sense pH over different ranges depending on their surface properties (Figure 5D). Electron spin longitudinal relaxation time, *T_1_*, of NV centers in a carboxylated FND uniformly varies depending on the external pH between pH 3 and pH 7, whereas there is no correlation between *T_1_* and pH range from 7 to 11. On the other hand, poly L-cysteine functionalized FND exhibits pH dependent *T_1_* changes between pH 7 and pH 11 [50]. Interestingly, temperature induces a broadening and a red-shift of the zero-phonon line (ZPL) of NV^−^. Using this unique phenomenon, FND-gold nanorod hybrids were developed as combined nanoheater/nanothermometers that generate local heat and enable measurement of the local temperature changes by recording the spectral shift of the zero-phonon lines of NV^−^ centers in human embryonic kidney (HEK) 293T cells under 594 nm laser irradiation. The high measurement sensitivity resulted in a detection limit of ± 2 °C/Hz^1/2^ [52]. Cellular redox processes have also been measured by quantum sensing techniques applied to functionalized FNDs [26]. NV^−^ centers in FND were coupled with nitroxide radicals in the presence of a bioinert PG polymer coating of the NDs. Due to the high sensitivity (detection limit: approximately 10^−23^ mol) of the *T_1_* spin relaxation time of the NV centers to the number of nitroxide radicals, the redox chemical process of ascorbic acid near the FND surface in an aqueous environment under ambient conditions could be sensed via changes in the *T_1_* spin relaxation time (Figure 5E). Sensing applications have also been developed based on the charge dynamics of the NV center, which can dynamically transition between NV^0^ and NV^−^. Owing to the large spectral shift between the ZPLs of the two NV charge states (NV^0^ at 575 nm and NV^−^ at 637 nm), the charge state can be readily distinguished optically. Distinct charge states can be modulated by surface charge from surface functional moieties or in the presence of charged molecules [15,74]. Based on these reversible charge state dynamics of NV centers, both pH and temperature were optically measured by changes of fluorescent spectra that reported on transitions between the two charge states [51]. The FND surface was modified with a polymer that changed its charge state as it swelled and collapsed in response to pH and temperature. By measuring the ratio of the spectral maximum to the NV^0^ ZPL pH could be measured over the range of 5.1 to 8, and temperature from 15 to 40 °C.

To increase sensitivity, expand sensing range, and to provide multifunctionality in sensing applications, surface modifications of FNDs are essential. For example, a gold nanorod (GNR) conjugated FND nanohybrid (GNR-FND) was prepared to generate dual nanoheater/nanothermometers [52]. To attach the gold nanorods to the FND surface, carboxylated FNDs were covalently functionalized with polyethylenimine (PEI) through standard carbodiimide chemistry, followed by decorating the cationic FND surface with citrate capped gold nanorods by self-assembly through electrostatic interactions. In a recent exciting development, surface modification of FNDs with a polycysteine layer has been shown to shift the pH sensing range of FNDs by increasing the p*K*_a_ of the surface groups [50]. While the *T_1_* of NV centers in carboxylated FND is sensitive to pH between pH 3 and pH 7, it shows no dependence on pH from 7 to 11. The pH detection of the NV centers was shifted to the pH range of 7 to 11 by introducing polycysteine, which contains a large number of thiol groups and therefore a higher p*K*_a_, onto the surface of the NDs. This proof of principle demonstration suggests a promising approach to enhance or modify the sensing capabilities of FNDs via selective surface modification approaches. In an another innovative approach to enhance the sensing capabilities of FNDs, they were coated with a thermo-responsive hydrogel shell conjugated with magnetic nanoparticles (MNPs). Taking advantage of the stimulus-responsive hydrogel as the transducer, the hydrogel grafted FND allowed ODMR-based temperature measurements [143]. Since the magnetic field at the FND from the MNPs decreases sharply with distance, the ODMR signal exhibited a steep dependence on the volume phase transition of the hydrogel that altered the distance between the FND and the embedded MNPs.

## 7. Conclusions

FNDs offer the possibility of being an ideal nanoparticle for numerous applications due to their exceptional physicochemical properties. In particular, the unique features of the NV^−^ center such as magnetic (spin triplet) ground state and the dependence of its fluorescence intensity on the spin state offer one of the most sensitive and attractive nanoscale sensing applications for magnetic and electric fields, in addition to temperature. To implement the potential applications of FNDs, suitable FND surface functionalization is required depending on the specific applications. Recently, smart surface functionalization strategies using external stimulus-responsive materials have been shown to increase sensitivity, detection range, and functionality of FND-based sensors. Consequently, rational, simple, and reproducible surface modification strategies using specific functional materials and structures should provide much higher efficiency, sensitivity, and broad sensing range of FND-based sensors.

Numerous functionalization methods, starting with homogenization of the ND surface to facilitate subsequent grafting reactions and conjugation of specific molecules, have been developed and continue to be refined and improved. Although several different surface moieties can be formed on NDs via specialized treatments, COOH and OH groups are widely used as versatile starting groups for the conjugation of functional moieties due to the possibility of a broad spectrum of additional reactions, reliability, and reproducibility. The conjugation reaction of specific molecules can be achieved via direct covalent or non-covalent reactions (physical adsorption and/or encapsulation). However, each approach has inherent strengths and weaknesses, some of which may impact the sensing and detection application. Despite the simplicity and flexibility of non-covalent functionalization, the activity or availability of physically adsorbed molecules on ND surface can be degraded due to structural changes or unfavorable orientations of the molecules. In addition, aminofunctional trialkoxysilanes on silica surfaces tend to lose their activity by internal hydrogen bonding between amine groups and surface silanol groups [148]. The presence of alkyl groups on silica surfaces have been shown to induce a hydrolytic lability resulting in reduced colloidal stability [109]. Therefore, multiple reliable, reproducible, rational, and appropriate surface functionalization strategies of NDs need to be considered, and thoroughly tested and verified, for specific applications.

## Figures and Tables

**Figure 1 nanomaterials-11-00153-f001:**
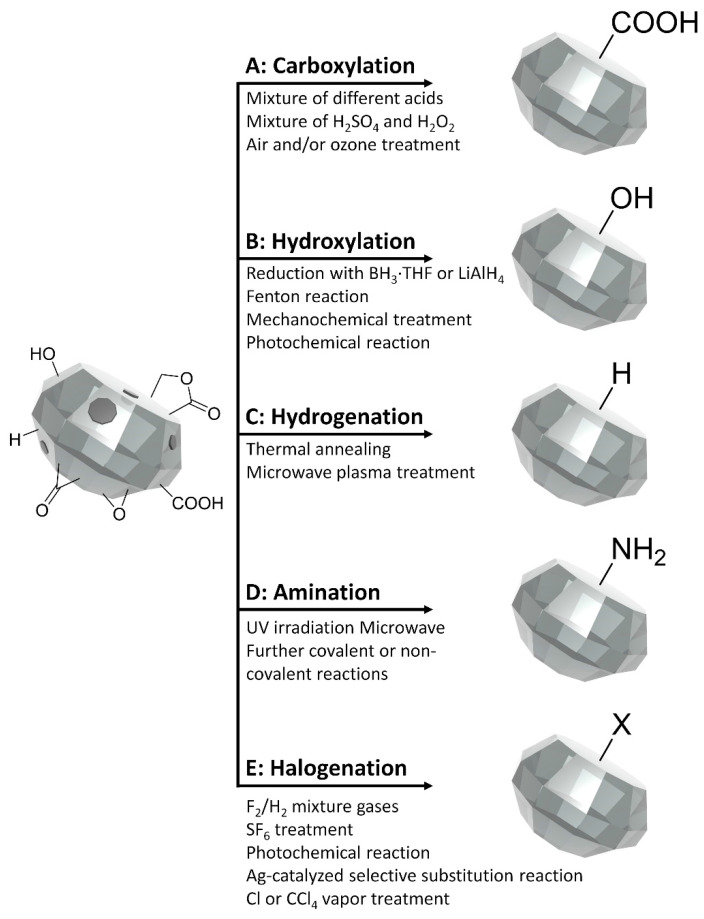
Homogenization of ND surface chemistry. The ND surface comprises different groups including sp^2^ carbon structures (dark gray), carboxylic acid, hydroxyl groups, lactones, ketones and ethers (left). The actual surface moieties and their relative abundances are dictated by the production methods and subsequent purification steps. ND surface homogenization schemes include: (**A**) Carboxylation via oxidation, (**B**) Hydroxylation via four different strategies, (**C**) Hydrogenation via thermal annealing or microwave plasma treatment, (**D**) Amination, and (**E**) Halogenation via several different approaches.

**Figure 2 nanomaterials-11-00153-f002:**
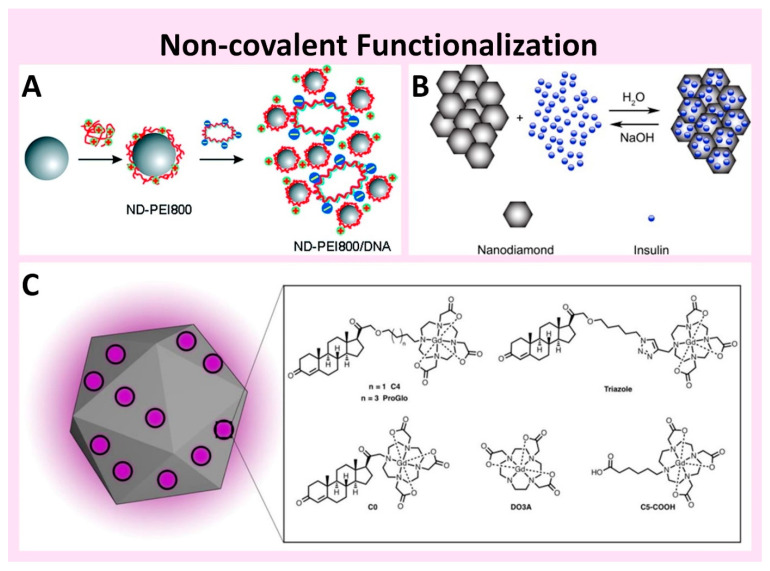
Non-covalent physical adsorption functionalization approaches: (**A**) Positively charged 800 Da polyethyleneimine is immobilized on the negative charged ND surface via electrostatic interactions. Reproduced from [78], with permission from American Chemical Society, 2009; (**B**) Hormone insulin adsorbed onto the ND surface through electrostatic interactions and H-bonding is used as drug delivery vehicle. Reproduced from [79], with permission from Elsevier, 2009; (**C**) To increase the aqueous solubility of the steroid receptor-targeted magnetic resonance imaging contrast agent ProGlo (purple circles), it was physically adsorbed to the ND surface. Due to the increased solubility, the resulting ND-ProGlo particles were efficiently accumulated in T47D breast cancer cells expressing progesterone receptors. Reproduced from [80], with permission from American Chemical Society, 2019.

**Figure 3 nanomaterials-11-00153-f003:**
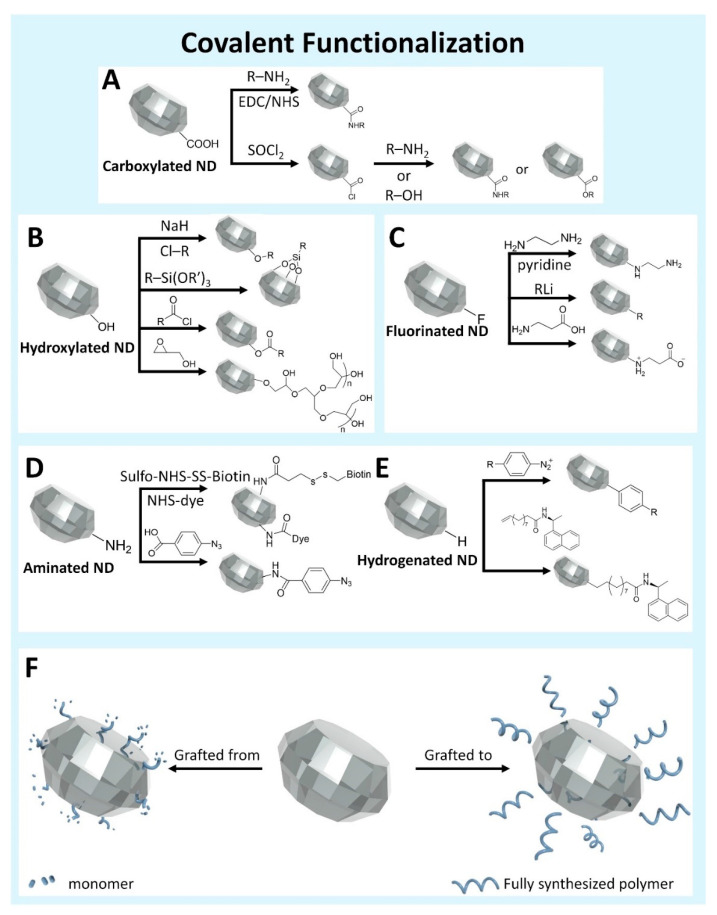
A variety of functionalization methods can be achieved through covalent surface modification of NDs surface based on the initial surface group on the homogenized NDs surface [58,81,82,83,84,85]. (**A**) Carboxylic acid coupling to amines or alcohols. (**B**) Hydroxyl group initiated functionalization. (**C**) Substitution based coupling with halogenated NDs. (**D**) Amine coupling to NHS esters or Carboxylic acids. (**E**) Generation of robust carbon–carbon single bonds from hydrogenated ND. (**F**) Polymer coating by direct polymerization reaction on ND surface (left, grafted from), or coupling of fully synthesized polymer to ND surface (right, grafted to) [86,87].

**Figure 4 nanomaterials-11-00153-f004:**
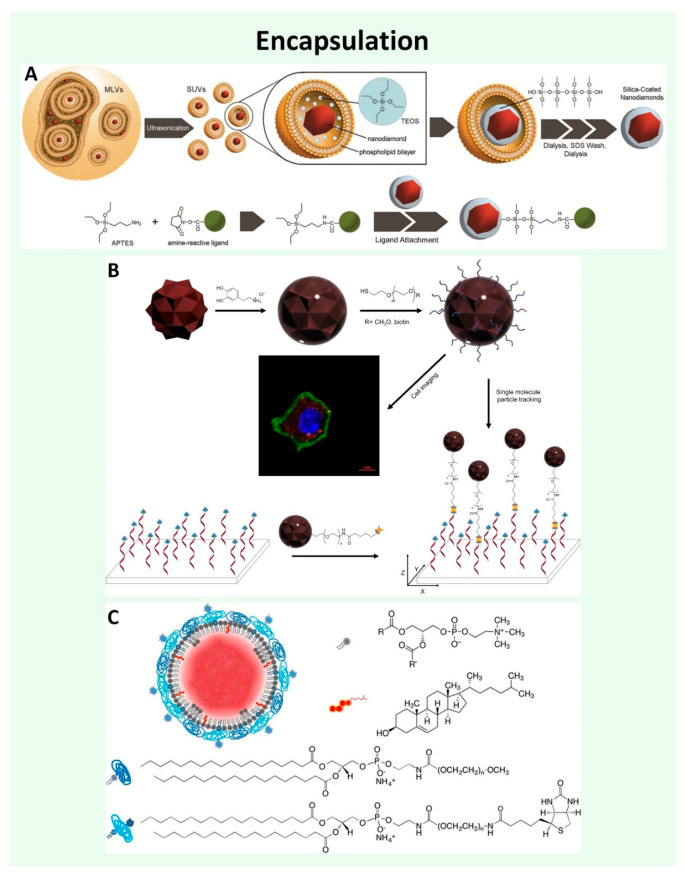
Encapsulation approaches for ND functionalization. (**A**) Silica encapsulation. Multilamellar 1-palmitoyl-2-oleoyl-*sn*-glycero-3-phosphocholine vesicles are ultrasonicated to form small unilamellar vesicles containing nanodiamonds (ND) with tetraethyl orthosilicate (TEOS). TEOS forms into a silica shell on the NDs in the presence of triethylamine as base catalyst. The silica surface can be further functionalized with APTES coupled to biotin (green sphere) via an amine reactive ligand; (**B**) Polydopamine encapsulation. Dopamine readily forms polydopamine (PDA) via self-polymerization reaction under basic solution conditions and the resulting PDA deposits onto the ND surface. Catechol/quinone groups on the PDA shell allows secondary reactions with amine or thiol containing molecules via Schiff base formation or Michael addition, respectively. PDA encapsulated NDs are functionalized with PEG-thiol for cellular internalization studies, or with heterobifunctional biotin-PEG-thiol to couple the PDAFNDs to DNA molecules (red helices) via streptavidin (yellow star) for single-molecule imaging and tracking; (**C**) Lipid encapsulation. NDs are spontaneously encapsulated with a mixture of L-α-Phosphatidylcholine from chicken eggs (Egg PC, grey top), cholesterol (red, second from top), and 1,2-distearoyl-*sn*-glycero-3-phosphorylethanolamine (DSPE) coupled to PEG (DSPE-PEG2000, grey and dark blue, second from bottom) or PEG-Biotin (DSPE-PEG2000-biotin, bottom). Lipid encapsulation stabilized the ND, whereas the biotin permitted subsequent labeling with functionalized neutravidin; (**A**) Reproduced from [123], with permission from American Chemical Society, 2013; (**B**) Reproduced from [110], with permission from John Wiley and Sons, 2018; (**C**) Reproduced from [130], with permission from American Chemical Society, 2018.

**Figure 5 nanomaterials-11-00153-f005:**
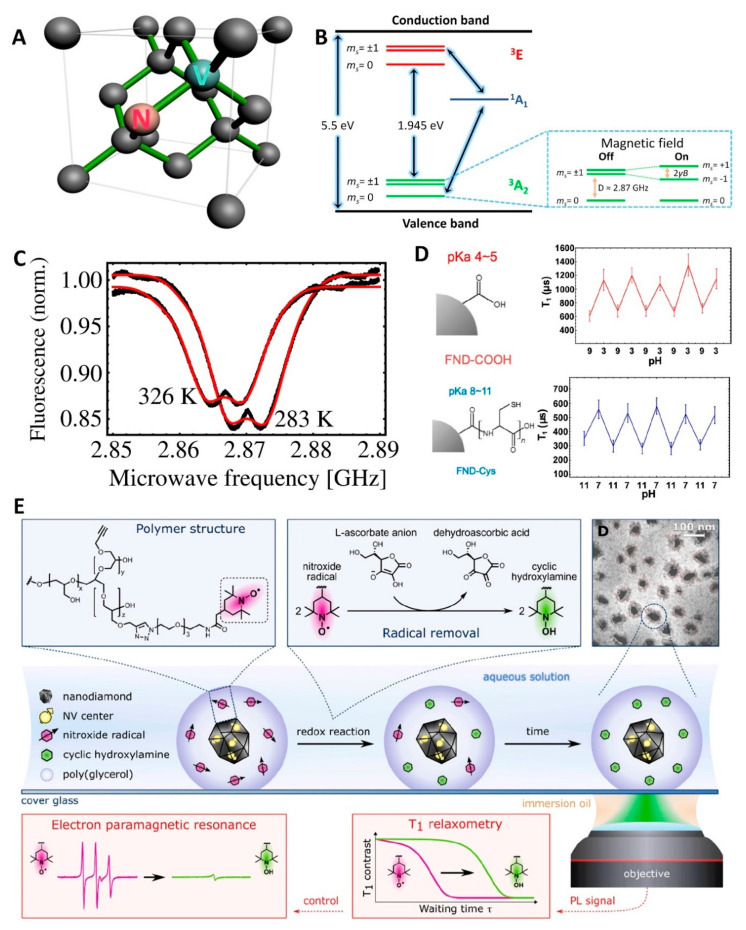
Characteristics of NV^−^ centers and their sensing applications. (**A**) Crystallographic structure of an NV center in diamond, consisting of a substitutional nitrogen (orange) adjacent to a vacancy (green); (**B**) Energy-level diagram of an NV^−^ center. A 1.945 eV spin conserving optical transition is required to excite the NV^−^ from the ground-state (^3^A_2_) to the excited-state (^3^E) triplets. Transitions from the *m_s_* = ± 1 sublevels of ^3^E to an intermediate spin singlet (^1^A_1_), which decays to the *m_s_* = 0 ground state, are much stronger than those from the *m_s_* = 0 sublevel. The spin-selective nature of this decay path can be used in conjunction with the 1.945 eV transition to optically pump the system into the *m_s_* = 0 state. The expanded blue box shows the three spin sublevels with *m_s_* = 0 and *m_s_* = ± 1 in zero and nonzero magnetic fields, D is the zero-field splitting, and 2*γB* is the Zeeman splitting, where *γ* is the electron gyromagnetic ratio; (**C**) Thermal shifts of the spin resonance of NV^−^ centers in FND at 283 and 326 K. The splitting observed in the ODMR spectra are due to crystal strain. Reproduced with permission from [18]; (**D**) The *T_1_* relaxation time can be modulated by changes in pH around the p*K*_a_ of surface groups. Reprinted from [50], with permission from American Chemical Society, 2019; (**E**) Rational Functionalization scheme of FND using nitroxide radical conjugated PG polymer shell for spatiotemporal readout of a redox chemical process. Reprinted from [26], with permission from American Chemical Society, 2020.

## Data Availability

The data presented in this study are available on request from the corresponding author.

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
