# Peer review of "Surface Modification of Fluorescent Nanodiamonds for Biological Applications"

_nanomaterials, 2021, doi:10.3390/nano11010153_

Round 1
Reviewer 1 Report
Wow, great review!
The review is providing on overview of surface modification and sensing application of FNDs with the special focus to biotechnological applications. The review is very much needed by the scientific community working with FNDs, due to the sum-up of all relevant functionalization strategies, the critical evaluation, and the great illustration. To have that summery on just 10 pages (excluding figures and references) is doubtless a big advantage for the future work of others.
Also, the shorter part to the N-V-centers and the related sensing applications strengthen the paper.
From my point of view the review fits exactly to the scope of nanomaterials and delivers a very strong impact to the scientific community. I recommend publishing of the article.
Author Response
We thank the reviewer for their interest and their positive evaluation of the review.
Reviewer 2 Report
Authors present a review paper about Fluorescent nanodiamond (FNDs). This review provides an overview of nanodiamond surface functionalization methods including homogeneous surface termination approaches (hydrogenation, halogenation, amination, oxidation, and reduction), in addition to covalent and noncovalent surface modification with different functional moieties. Morover authors presented the biomedical applications of nanodiamonds are discussed in the context of functionalization. This work is presented legibly and clearly.
Author Response
We thank the reviewer for their positive comments and evaluation of the review.
Reviewer 3 Report
There are lots of similar reviews, which are better organized and give more information to the readers. Moreover, Authors do not cite them. Some examples are given in the attached file.
The title is misleading. Authors describe general methods of NDs functionalization. There is only one chapter devoted to fluorescent NDs - the 6th one.
My general impression is, that the Authors do not have a concept for this paper. It is just the summary of information, without any discussion and meaningful conclusions.
Please find more remarks in the attached file.

Author Response
We thank the reviewer for their critical reading of the manuscript and their constructive comments. We address the reviewer's comments in detail below and have revised the manuscript with these specific and general critiques in mind.
- In page 1 at the title;
Similar reviews: Surface functionalization of nanodiamonds for biomedical applications https://doi.org/10.1016/j.msec.2020.110996 NOT CITED!
And more...
K Turcheniuk and Vadym N Mochalin 2017 Nanotechnology 28 252001 NOT CITED!
Nanodiamonds with powerful ability for drug delivery and biomedical applications: Recent updates on in vivo study and patents. https://doi.org/10.1016/j.jpha.2019.09.003 NOT CITED!
We thank the reviewer for recommending these references. We added these citations in the introduction and section 6 as references 4 to 6.
- In page 1 at the title; Not all of nanodiamonds are fluorescent, not all of the references given in the manuscript describe fluorescent nanodiamonds.
The reviewer raises a valid point concerning the fact that the cited literature covers functionalization of both nanodiamonds (NDs) and fluorescent nanodiamond (FNDs). Most of the surface modification strategies of ND have been developed for detonation nanodiamonds (DNDs) due to the early development of this synthetic method, and the fact that they are commercially available. Our goal is to provide the reader with an expansive overview of ND functionalization approaches, some of which have been demonstrated with FNDs, but the vast majority if not all of the functionalization approaches could be extended for use with FNDs. Due to the increasing development of FNDs for a variety of applications, most notably in the biomedical fields, a review that presents FND functionalization within the broader context of ND functionalization is a useful addition to the field since a significant fraction, if not the majority, of the field is not well-versed in the broader ND functionalization literature, which is nonetheless germane and important for the FND field.
- In page 1 at the abstract; surface of diamond is inert in contras to nanodiamonds This senstence does not make sense.
We thank the reviewer for pointing out this potential source of confusion. We are simply drawing attention to widely held notion that diamond surface is completely inert. We have revised the sentence to make the general point concerning the surface groups and subsequent functionalization of diamond in general, without make the implied assertion that only nanodiamonds can be functionalized. “Although diamonds are generally considered inert, they typically possess diverse surface groups that permit a range of different functionalization strategies”.
- In page 1 at the abstract; homogeneous surface termination. I do not understand this phrase. What homogenous means in this context?
The reviewer raises a good point concerning the homogeneous surface functional groups of FND. As we mentioned in section 2, pristine NDs have various surface moieties including carboxyl, hydroxyl, carbonyl, different alcohol, and ether groups as well as nondiamond carbon structures and impurities that are generated during synthesis. Diverse surface moieties and impurities will diminish the reproducibility of functionalization of the NDs. Thus, in this context, homogenization means creating uniform surface functional groups.
- In page 1 at the abstract; Furthermore, the subsequent coupling of biomolecules onto functionalized nanodiamonds is reviewed. I see no examples in the manuscript of biomolecules bonding to FNDs surface.
We included examples of non-covalent attachment of bovine insulin, apoobelin, lysozyme, and doxorubicin hydrochloride. Although we focused on the chemical reactions for functionalization of NDs via covalent bonds, most of citations includes conjugation schemes with different molecules such as FITC-streptavidin, N,O-carboxymethyl chitosan, anticancer drugs, DNA, polymers, peptides, and other biomolecules. Please see the citations in sections covering non-covalent functionalization and the combination of non-covalent and covalent functionalization.
- In page 1 at the abstract; Finally, biomedical applications of nanodiamonds are discussed in the context of functionalization. It is not true. Only few examples of using FNDs in sensing are given. There is no discussion on the functionalization and its influence of FNDs properties.
We agree with the reviewer that the primary focus of the applications section was on sensing applications of FND. We have accordingly added several recent examples of biomedical application of FNDs in the revised manuscript. (please see section 6)
- In page 1 at introduction; Nitrogen-vacancy (NV) centers. For sombedy who is not familiar with FNDs it is not clear why nitrogen is present in the structure.
The reviewer raises a valid point concerning the presence of nitrogen in the diamond structure. To address this potential source of confusion, we added a sentence “ Natural diamonds commonly contain nitrogen atom impurities, and nitrogen is frequently doped into synthetic diamonds.” (line 31-33).
- In page 2 at introduction; phenomena has been. grammar.
We thank the reviewer for pointing out this error, which we have corrected in the revised manuscript. phenomena → phenomenon (line 58).
- In page 2 at introduction; Owing to accommodation of other color centers such as N3, H3, SiV, GeV, and Ni. how this centers are created?
We appreciate the reviewer pointing out that we have not explained the creation process of any of the center, but we have focused more on the functionalization schemes of NDs and FNDs. A detailed explanation of the synthesis approaches of the various centers is beyond the scope of this review. We have included the relevant information in citations 20 to 23 (line 63).
- In page 2 at introduction; fac.
We thank the reviewer for pointing out this typo. We have corrected this in the revised manuscript (line 93).
- In page 2 at introduction; Therefore, many of the functionalization strategies originally developed for DNDs have been successfully adapted for use with HPHT nanodiamonds. references should be added.
We thank the reviewer for pointing out this oversight and have added citations to the relevant literature in the revised manuscript (line 107).
- In page 3 at section 2; Homogenization of surface moiety of NDs. I do not understand this phrase. In my opinion "homogenization" is used not correcty here. to make uniform is a proper term
We appreciate the reviewer raising this potential source of confusion. We have clarified the homogenization terminology, but we point out that it is in common usage in the nanodiamond and surface modification literature more generally (line 119).
- In page 3 at section 2. Formation of uniform surface moiety of NDs; Pristine NDs have various surface moieties as well as nondiamond carbon structures and impurities that are generated during synthesis. All paragraphs describe NDs functionalization not FNDs.
Whereas the reviewer is correct that most of the descriptions are NDs functionalization, FND is one of the ND. NDs are typically classed as diamond particles less than 100 nm in size. It means NDs include FND and DND. In addition, formation of uniform surface moiety and surface functionalization scheme of DND can be applied to FND. Consequently, we use a “NDs” instead of FND to cover both DND and FND. We have included a short clarification of the distinction between FND and ND in the introduction to address the more general concerns related to discussion of ND versus FDN functionalization schemes.
- In page 3 at section 2. Formation of uniform surface moiety of NDs; is carried out four different ways. grammar.
We think that the grammar is correct – we are describing 4 different hydroxylation approaches (line 148).
- In page 4 at section 2. Formation of uniform surface moiety of NDs; The Fenton reaction also presents. presents?
We replaced “presents” with generates to eliminate potential confusion with this sentence (line 156).
- In page 4 at section 3. Non-covalent functionalization; Positive charged. grammar.
We thank the reviewer for pointing out this error that we have corrected in the revised manuscript.
- In page 8 at section 5. Combination of non-covalent and covalent functionalization; approaches compensates for disadvantages. grammar.
We thank the reviewer for pointing out this oversight. We changed the “compensates” to “compensate” (line 364).
- In page 8~9 at section 5. Combination of non-covalent and covalent functionalization; gown.
We thank the reviewer for pointing out this error. We changed “gown” to “grown” (line 385~386).
- In page 9 at section 6. Sensing applications of FNDs; The properties of NV― centers within FNDs are promising nanosensors. sensor is a device.
The reviewer raises an excellent point concerning the meaning of nanosensor. We changed this sentence to “The properties of NV― centers within FNDs are promising for sensing applications” (line 457).
- In page 9 at section 6. Sensing applications of FNDs; In addition to ODMR, different sensing methods including spin coherence, spin relaxation time, spectral shift of the zero-phonon lines, and charge-state have been proposed to measure the concentration of paramagnetic ions and molecules, temperature, and pH. It should be added that FNDs were used.
The reviewer raises a valid point concerning the use of FNDs for these sensing applications. We modified the sentence appropriately (line 476).
- In page 11 at Conclusions; FNDs offer the possibility of being an ideal nanoparticle for numerous applications due to their exceptional physicochemical properties. there are no examples in the paper proving this statement.
Although the statement is described a fact, we agree with the reviewer that primarily sensing applications of FND are presented in the review. We have added several biomedical applications of FND in the revised version that relate to this statement.
- In page 11 at Conclusions; Recently, smart surface functionalization strategies using external stimulus-responsive materials have been shown to increase sensitivity, detection range, and functionality of the FNDs. how FNDs can have sensitivity or detection range?
The reviewer raises an excellent point concerning the sensitivity and detection range of FNDs. We modified the text to specify the detection limit of the FND-based sensing applications in the revised manuscript on pages 11.
- In page 11 at Conclusions; using specific functional materials and structures should provide much higher efficiency, sensitivity, and broad sensing range to the sensing applications of NDs. sensitivity and sensing range can have a sensor not NDs or their applications.
We thank the reviewer for pointing out the subject of sensitivity and sensing range of FNDs, when we should be referring to the sensing method based on the FNDs. We have modified the text accordingly.
We thank the reviewer for the very careful reading of the manuscript and the insightful comments and corrections. The review has benefited substantially due to the efforts of the reviewer.
Reviewer 4 Report
io-medical application of fluorescent nano-diamond particles.
This well written review may be interesting and helpful for a wide reader auditorium.
Only few remarks requiring minor revisions may be mentioned:
1. While most of references in the manuscript are appropriate and adequate it seems strange to see ref. [4] Davies, G.; Hamer, M.F.; Price William, C. Optical studies of the 1.945 eV vibronic band in diamond. Proc. R. Soc. London Ser. A-Math. Phys. Eng. Sci. 1976, 348, 285-298.
This reference to almost half century old publication does not belong definitely to signficant pioneering work and should be replaced by one of recent papers. Please, consider, e.g., DOI: 10.1002/pssb.201700189 and DOI: 10.1016/j.optmat.2017.10.019 reporting comprehensively about different color centers in nano-sized diamond.
2. There are some need in careful reading of the text for misprint corrections.
Author Response
We thank the reviewer for pointing this oversight on our part and for recommending the more appropriate recent citations. We replaced the citation with the recommended citations, currently numbers 9 and 10 in the revised manuscript (line 41).
Round 2
Reviewer 3 Report
I have already gave my opionon and I do not want to judge on the manuscript.